# Fostering Nursing Staff Competence in Personal Protective Equipment Education during COVID-19: A Mobile-Video Online Learning Approach

**DOI:** 10.3390/ijerph19159238

**Published:** 2022-07-28

**Authors:** Hsiu-Ju Jen, Kuei-Ru Chou, Ching-Yi Chang

**Affiliations:** 1Department of Nursing, Taipei Medical University-Shuang Ho Hospital, New Taipei 235, Taiwan; 15033@s.tmu.edu.tw (H.-J.J.); kueiru@tmu.edu.tw (K.-R.C.); 2School of Nursing, College of Nursing, Taipei Medical University, 250 Wuxing Street, Taipei 11031, Taiwan; 3Center for Nursing and Healthcare Research in Clinical Practice Application, Wan Fang Hospital, Taipei Medical University, Taipei 11031, Taiwan; 4Psychiatric Research Center, Taipei Medical University Hospital, Taipei 11031, Taiwan; 5Neuroscience Research Center, Taipei Medical University, Taipei 11031, Taiwan

**Keywords:** nursing, nurses’ education, mobile video online learning approach, personal protective equipment

## Abstract

Nursing staff who are competent to use personal protective equipment (PPE) correctly can protect themselves while providing safe, high-quality care to patients. Under pandemic conditions, the ability to wear PPE correctly is essential in clinical practice, but the acquisition of correct PPE-wearing procedures is difficult for most staff in the absence of live practice drills. This study aimed to test the mobile video online learning approach by integrating PPE contexts into a digital learning system. We conducted an experiment to verify whether the mobile video online learning approach could effectively improve nursing staff’s learning achievement, learning anxiety, critical thinking skills, and learning self-efficacy. The study used a quasi-experimental design and was conducted with 47 nursing staff, divided into one group using a mobile video online learning approach and one group with a conventional learning approach. We used pre-and post-test examinations of learning achievements, learning anxiety, critical thinking, and learning self-efficacy. Results showed a significant effect of using the mobile video online learning method in helping nursing staff to decrease learning anxiety and improve knowledge about COVID-19 protection, increase learning achievement, critical thinking skills, and learning self-efficacy. These benefits are of interest to nursing workplace managers wishing to maintain professional standards during epidemics by improving the nursing staff’s PPE knowledge and self-efficacy concerning PPE.

## 1. Introduction

During the COVID-19 pandemic, ensuring both patient safety and nurse staff work safety are of importance. Social distancing measures introduced to mitigate COVID-19 transmission and its impact on higher education institutions and practical learning environments have created a wide range of challenges [1]. Due to COVID-19, educational institutions have been forced to switch rapidly from face-to-face teaching to online teaching, which has presented a challenge to existing educational institutions and managers. The provision of digital teaching was immediately implemented, and learners have been forced to adapt to an online education dissemination system on a large scale [2]. The online environment, digital learning pedagogy, preparation of teaching equipment, and information literacy of learners all present challenges for teachers to overcome [3]. Some learners have readily accepted the shift to digital learning, with its associated greater flexibility in terms of location and time, while others have experienced discomfort and anxiety because of their limited information literacy or lack thereof [4].

Lonsdale et al. [5] contend that with the mobile video online learning approach (MVOLA), teachers are better able to design classroom-led activities that allow learners to engage actively with the material, apply constructive knowledge, and discuss and practice the material with their peers. As a result, guiding learners to practice self-disciplined learning and discussion via digital platforms has become a learning model that is often adopted by schools [6]. Zou, Luo, Xie, and Hwang [7] pointed out that for high-quality classroom learning, practice activities are essential; however, researchers have also found that without proper learning design and support, the MVOLA may reduce learner engagement and increase learning load [8]. A retrospective article found that an effective MVOLA can motivate learners to learn actively from video lectures and through self-guided learning [9].

The disruptive impact of COVID-19 has necessitated a rapid change in clinical practice. For example, the use of video consultations in the UK has now been implemented at a speed and scale that are in stark contrast with the extremely slow rate of implementation characteristic of the pre-COVID era [1]. Thus, to prevent any interruption of participant learning, nursing education has shifted to digital learning, replicating traditional classroom activities via online synchronous or asynchronous lectures and providing a flexible learner-centered learning environment [10]. A retrospective article by Chang, Lai, and Hwang [11] showed that digital learning produces positive academic outcomes in nursing and health professional education, ensuring that nursing staff learn effectively and facilitating nursing staff–peer engagement when learning about clinical practice.

With these changes in pedagogy and education, the COVID-19 pandemic has permanently altered curriculum content. Even the most experienced practitioners face unprecedented challenges. Managers urgently need to help nursing staff to understand the ever-changing information about how to protect people from COVID-19 infection, safely care for patients, and educate their families [12]. The nursing staff members have to develop therapeutic relationships with their anxious patients while wearing multiple layers of personal protective equipment (PPE) and communicating with patients’ families over the phone. Promoting a better understanding of critical care and nursing care among frontline healthcare workers is a major element of infection prevention and control [13]. This includes knowledge about PPE, the timing of handwashing, and infection control principles. Ensuring that medical staff members are protected means more effective containment of the virus [14]. To prevent COVID-19 from causing further pandemics in the future, it is crucial to teach nursing staff to correctly identify and implement the protective measures required to reduce the risk of infection, and to cultivate their self-confidence and core competencies in dealing with COVID-19 [15].

The devastating impact of COVID-19 presents an unprecedented transformational opportunity for nursing education. The rapid dissemination of innovations in evidence-based education to support the clinical nursing practicum in response to COVID-19 and beyond is imperative. It is crucial for nurse managers and researchers to train nursing staff and their clinical partners together to properly oversee, assess, and communicate such transitions. PPE has been developed for nurses to use in the COVID-19 pandemic, and it is crucial that nurses use it when they engage with patients [16]. The several waves of the COVID-19 pandemic have had a substantial impact on healthcare and education, and nurses have played a critical role in responding to the pandemic [17]. Cooperation between educational and clinical institutes when providing training has helped to ensure the effectiveness of this response. In earlier studies, the correct use of PPE was established based on standard protocols [18,19]. The correct use of PPE has been highlighted as important for mitigating the waves of the pandemic and essential for limiting and responding to the prevalence of infections (Iheduru-Anderson, 2021) [20]. Knowledge of how to use PPE correctly has been essential in the healthcare system, and in clinical settings particularly, during the COVID-19 pandemic [21].

Recently, investigators have conducted studies on the use of technology to improve learners’ ability to use PPE correctly for the protection of themselves and their patients [22,23]. Several studies have investigated the effectiveness of videos for conveying information related to COVID-19 [24]. In this context, teaching videos dealing with COVID-19 PPE can improve donning (putting on) and doffing (taking off) competence and self-efficacy [24,25], although this competence and self-efficacy can be difficult to teach thoroughly in a classroom setting. Few studies have applied the MVOLA to PPE educational activities in the nursing arena, however. This study uses the MVOLA in digital PPE instruction that is designed to guide learners in conducting repeated practice and observation and critical thinking, enabling them to reflect on the use of PPE and their knowledge of COVID-19; therefore, the purpose of this study is that nursing staff simulate PPE use while being instructed via the MVOLA, their ability to use PPE correctly, learning achievement, learning anxiety, critical thinking skills, and learning self-efficacy in donning (putting on) and doffing (taking off) PPE will improve.

## 2. Aim

This study aimed to test the MVOLA by integrating PPE contexts into a digital learning system. We conducted an experiment to verify whether MVOLA could effectively improve nursing staff’s learning achievement, learning anxiety, critical thinking skills, and learning self-efficacy.

## 3. Methods

### 3.1. Study Design

A quasi-experimental approach was used to examine the effectiveness of the MVOLA as a component of the PPE nursing education course. To investigate the effectiveness of the proposed approach, the learning achievement, critical thinking, learning self-efficacy, and learning anxiety of participants were investigated with a pre-and post-test quasi-experimental design.

### 3.2. Study Settings and Participants

A total of 47 nurses volunteered to participate in this study after being recruited by mail. They were randomly assigned to a control group of 23 participants and an experimental group of 24 participants. This sample size was verified with a post hoc power analysis using G*Power 3.1, which yielded a statistical power of 0.80. The study was reviewed and approved by the university’s research ethics committee. Before the experiment, a consent form was signed by the participants, who were allowed to withdraw at any time without affecting their daily work or rights.

### 3.3. Intervention

The ability to use PPE correctly and knowledge of how to protect against disease transmission have become important competencies for nursing staff to attain before they are enrolled to work in a real-world clinical situation. We designed an MVOLA for teaching content about COVID-19-targeted PPE and protection against disease transmission. The content was implemented on an e-learning platform and incorporated a new feature to help nursing staff address a critical daily challenge during the pandemic. The 20-min course demonstrates the proper techniques and sequence for donning and doffing PPE, and the nursing staff was able to repeat the learning modules at any time or location using their smartphones. We also carried out an experiment that involved dividing nursing staff into two groups. The experimental group used the mobile video learning method, while the control group used the lecture learning method with two-dimensional images in PowerPoint as visual aids. Furthermore, we assessed the effect of this approach on the nursing staff’s learning anxiety, critical thinking, learning self-efficacy, and PPE-related performance.

### 3.4. Instruments

The instruments employed in the study consisted of a pre-test, a post-test, and the measuring tools used to assess learning achievement, learning anxiety, critical thinking, learning self-efficacy, and demographic data. All data collection was conducted on Google Forms. Four measuring instruments were used to assess the learners’ learning performance and affect: a multiple-choice quiz to measure knowledge of PPE procedures, a scale to assess learning anxiety, a questionnaire to evaluate critical-thinking ability, and a questionnaire to evaluate learning self-efficacy. The quiz questions were designed by two teachers of clinical nursing with more than 10 years of experience each to assess basic knowledge of PPE. The test consisted of 10 multiple choice questions and had a maximum score of 100 points.

The learning-anxiety scale was modified from the subtopic performance anxiety scale proposed by Thompson and Lee [26], and included 13 items with a five-point Likert scale and a Cronbach’s α value of 0.75. The questionnaire to evaluate critical thinking ability was revised based on the questionnaire proposed by Chai et al. [27] and included six items with a five-point Likert scale and a Cronbach’s α value of 0.85. The learning self-efficacy measurement tool was a questionnaire constructed by Pintrich, Smith, Garcia, and McKeachie, [28] with a total of eight items with a five-point Likert scale and a Cronbach’s α value of 0.80. One of the question statements was “I’m certain I can master the skills being taught in this class”.

### 3.5. Statistical Methods

The collected data were analyzed using SPSS software (IBM, SPSS Inc., Chicago, IL, USA). We used various descriptive statistics to present data and employed independent samples *t*-tests to analyze outcome metrics.

### 3.6. Study Process

The study process included two-phase. In the first phase, the two groups took a nursing education course as part of their vocational training before they entered the clinical workplace. This course included a two-hour module on PPE and hand hygiene skills, received via the MVOLA for the experimental group and via a lecture and PowerPoint for the control group. Subsequently, a pre-test of learning achievement, critical thinking, learning self-efficacy, and learning anxiety was administered for a period of 30 min.

In the second phase, the control group learned using the traditional learning method, in which the learners were asked to find the correct procedure for PPE and then work on the learning sheet using PowerPoint. The experimental group used the MVOLA, in which each participant was asked to search for PPE using the mobile video online learning system on their mobile devices. After the learning activities, all nurses completed the post-test and questionnaires.

## 4. Results

The demographic characteristics of the nursing staff are shown in Table 1. Demographic characteristics not specifically shown were comparable between the experimental and control groups.

The average age of participants in the experimental group was 24 years (SD = 0.79 years), and the majority were female (female: *n* = 21, 91.3%; male: *n* = 2, 8.7%). In terms of nursing hospital experience, 13 nursing staff (57%) had more than 1 year, 8 nursing staff (35%) had 6–12 months, and 2 nursing staff (13.7%) had 3–6 months. The majority of the nursing staff were unmarried (unmarried: *n* = 22, 95.7%; married: *n* = 1, 4.3%), and most had junior college level education (junior college: *n* = 17, 73.9%; university: *n* = 5, 21.7%; graduate school: *n* = 1, 4.3%).

In the control group, the average age of participants was 24 years (SD = 1.96 years), and the majority were female (female: *n* = 23, 95.8%; male: *n* = 1, 4.2%). In terms of nursing hospital experience, 14 nursing staff (58%) had more than 1 year, 9 nursing staff (38%) had 6–12 months, and 1 person (4%) had 3–6 months. The majority were unmarried (unmarried: *n* = 22, 91.7%; married: *n* = 2, 8.3%), and again most had junior college education (junior college: *n* = 16, 66.7%; university: *n* = 6, 25%; graduate school: *n* = 2, 8.3%).

Table 2 shows the independent samples *t*-tests of each of the factors assessed (learning achievement, learning anxiety, critical thinking, and learning self-efficacy) revealed no significant differences in the pre-learning status of any of the factors between the two groups (*t* = 1.03–4.45, *p* > 0.05). In contrast, the scores of the experimental group were significantly higher than those of the control group for all four factors after they had received their training (*t* = 1.57–1.87, *p* < 0.5). Thus, the MVOLA strategy significantly reduced the participants’ learning anxiety and improved their learning achievement, critical thinking, and learning self-efficacy.

## 5. Discussion

This study’s major finding is that implementation of an MVOLA for PPE education met students’ online learning needs, provided significantly greater learning effectiveness, and significantly reduced learning anxiety. The repeated practice and learning allowed the students to achieve their expected learning outcomes. Students should have the opportunity to develop and practice clinical competencies in a safe environment in order to protect themselves and their patients, reduce errors in donning and doffing PPE, and perform routine care in a clinical setting. Liu and Pei [29] pointed out that videos can help students to learn new skills, improve the learning process, and are considered to be effective teaching tools. Although few experimental studies have demonstrated their effectiveness for training in the use of PPE in the context of epidemics, our results show that using an MVOLA reduced the participants’ learning anxiety and improved their learning achievement, critical thinking skills, learning self-efficacy, and ability to don and doff PPE correctly.

Regarding learning achievement, the independent samples *t*-tests showed that the MVOLA yielded superior learning achievement outcomes compared to the traditional teaching approach. These findings echo those reported by Chiu et al. [30], who found that technological approaches using mobile learning activities can improve students’ learning outcomes. This suggests that this approach enables students to reflect during training sessions. The students’ metacognitive awareness drives them to build their procedural PPE knowledge and skills through mobile video online learning. Hwang, Wu, and Chang [31] showed that the mobile learning method is effective in helping to instantly improve personal learning expertise and skills. Students can use the WSQ approach to understand and complete learning tasks. A meta-analysis by Chen et al. [32] similarly found that mobile video online learning can stimulate learners to participate in clinical practice and enhance their learning outcomes in a co-working environment; specifically, the learning process delivered via videos can improve students’ knowledge and application ability. Several studies related to mobile learning have also pointed out that placing learners in specific contexts can help them to gain a better understanding of the learning content and identify tasks to learn [33]. This may also, to some extent, impact their decision-making ability in clinical practice; therefore, through repeated observations of PPE training material via mobile learning, students can acquire a deeper understanding of the informational content and improve both their ability to don and doff PPE correctly and their memory and understanding.

Regarding learning anxiety, our experimental results also demonstrated the effectiveness of the MVOLA in reducing PPE-related learning anxiety. This is assumed to be because the learners could repeatedly watch and review unfamiliar steps and areas in which they had any doubts about their knowledge. Some learners specifically mentioned that they had learning anxiety with respect to PPE. A possible reason for this is that PPE is a costly consumable that must be discarded after a single use. When resources are limited and there are many types of PPE—a common circumstance for nursing staff—these factors are especially important. Unfamiliarity with the donning process is common among nursing staff [34]. In the real-world clinical environment, a lack of familiarity with the required PPE-donning process leads to anxiety about caring for high-risk patients. The MVOLA, therefore, allows learners to develop confidence in the PPE donning and doffing procedure and in their knowledge about PPE and epidemic prevention, thereby enhancing their confidence in their ability to take care of high-risk patients safely. Research has also shown that learners’ educational level can significantly impact the effectiveness of mobile-learning technology [35]. As a result, identifying the best approaches to promoting mobile learning combined with technological applications in the workplace to target different medical practice groups requires further planning and discussion.

Regarding critical thinking, researchers have suggested that learners with better critical thinking skills tend to know how to identify gaps in their knowledge and select areas of expertise they need to improve on accordingly [36]. When learners learn by watching mobile videos, they can understand the entire PPE donning and doffing procedure and effectively complete their learning task without the necessity of interpreting abstract text-based descriptions. Researchers also contend that mobile video learning combined with scenarios makes learning safe and meaningful and mentally links the learner to real clinical situations, which improves learning outcomes [37,38]. According to previous studies, mobile video learning allows novice nurse practitioners to participate in classrooms in real time and provides an experience of the real world clinical environment, allowing them to integrate this new knowledge quickly into their daily care work [39,40]; therefore, integration of the MVOLA into the curriculum would effectively enhance ’students’ critical thinking, improve their awareness of the correct procedures for donning and doffing PPE, and enhance their ability to engage in professional daily care work.

Regarding learning self-efficacy, the MVOLA improved the participants’ learning self-efficacy compared to the traditional approach in this study, regardless of the participants’ education level. The low self-efficacy of the control group can be explained as the result of the absence of an audio–video introduction and the limited information that they received via reading, although this may vary among participants. The MVOLA provided participants with more opportunities to exchange ideas, leading to better understanding, and they were more confident in their ability to perform the correct PPE donning and doffing procedures, as shown by Peechapol et al. [41] and Kim and Suh, [42] who noted that participants’ learning self-efficacy could be improved by providing more opportunities to learn from others through mobile video online learning.

The main contribution of this study is that the MVOLA improved the students’ learning achievement, critical thinking, and learning self-efficacy with respect to donning and doffing PPE. It did so by developing their professional competencies, resulting in an ability to perform daily medical care tasks correctly. Involving learners in the PPE learning process can help to improve their ability to care for COVID-19 patients, take safety precautions, and translate their knowledge into an ability to perform routine clinical COVID-19 patient care tasks. Future research could be extended to training courses in other medical professions or schools and could assess the impact of the MVOLA on the development of student professional competencies in other fields. In addition, it could be combined with other learning strategies or learning assessment methods. For example, when applying the MVOLA, we should consider using it in conjunction with an objective structured clinical examination (OSCE) to verify its learning effectiveness and further analyze and compare its impact on learner performance.

## 6. Limitations and Recommendations

This study used data from only one institution, limiting the generalizability of the experimental results. In the clinical training environment, however, it is challenging to implement mobile video online learning effectively for new nurses. First, the epidemic situation presented numerous challenges, in addition to the fact that the training of nurses is already a difficult process. The monthly turnover of new nursing practitioners fluctuates greatly; thus, our sample size was limited; therefore, it is best to be conservative when generalizing these findings to other medical practitioners in training. Second, creating clinical medical and nursing teaching video clips, which requires human resources to create scripts, record, and edit material that is correct according to nursing standards, is another challenge. Medical nursing standards change in response to changes in the environment and new empirical discussions; therefore, teaching material must respond timeously to changing clinical practice needs. The proposed solution to these issues is to review and update clinical teaching materials every year. This would necessitate cooperation with digital experts and the further integration of teaching materials into e-learning platforms so that medical learning activities can achieve consistent results and quality of care.

## 7. Conclusions

Nurses caring for patients with COVID-19 are at high risk compared to those caring for general patients. The nursing work is challenging. Knowledge construction is essential for nursing staff, especially in the medical field. Such clinical work requires sufficient knowledge and critical thinking to minimize exposure to the risk of infection. Even allowing for the differences in the demographic characteristics of the learners in the two groups in our experiment, the effect of the intervention in the experimental group was significant. The recorded digital teaching materials were integrated into the online digital teaching platform. Mobile video online learning was implemented in the training and education of new clinical nurses to reduce their learning anxiety and effectively meet their learning needs. Allowing learners to use their personal devices and learn at their own pace is advisable because it means they can do so anywhere and as frequently as they wish without being restricted by their shift schedule. This education method enables learners to demonstrate professional environmental adaptability, respond more quickly to specific tasks, and learn complex tasks in a shorter time.

Our findings can help nursing managers to train nurses in PPE use, improve nurses’ critical thinking skills, reduce nurses’ PPE-related learning anxiety, improve their teaching strategies for building learning self-efficacy in PPE donning and doffing, and create a safe learning environment. Because of the COVID-19 pandemic, the shortage of nurses is currently even greater than under normal circumstances. It is critical to consider the fact that recently graduated nurses are required to care for patients with COVID-19; training, therefore, needs to be strengthened to ensure workplace safety. Considering the complexity of COVID-19 patient infection, it is necessary to introduce technological tools combined with digital education methods to rapidly make nurses aware of the latest epidemic-prevention knowledge, and thus reduce the inherent risks associated with the nursing care process. This study provides insight into the factors that affect the quality of nursing care in terms of ongoing professional development. It will also help to improve the quality of medical care overall. A nursing supervisor can emphasize the value of sustainable operation in strengthening the epidemic prevention knowledge of their nursing team and in professional training for medical safety.

## Figures and Tables

**Table 1 ijerph-19-09238-t001:** Participant demographic data.

Category	Experimental Group (*N* = 23)	Control Group (*N* = 24)
Age in years		
<25	16 (69.6%)	15 (62.5%)
25–30	7 (30.4%)	9 (37.5%)
Gender		
Female	21 (91.3%)	23 (95.8%)
Male	2 (8.7%)	1 (4.2%)
Duration of hospital experience		
3–6 months	2 (9%)	1 (4%)
6–12 months	8 (35%)	9 (38%)
>1 year	13 (57%)	14 (58%)
Marital status		
Unmarried	22 (95.7%)	22 (91.7%)
Married	1 (4.3%)	2 (8.3%)
Education level		
Junior college	17 (73.9%)	16 (66.7%)
University	5 (21.7%)	6 (25%)
Graduate school	1 (4.3%)	2 (8.3%)

**Table 2 ijerph-19-09238-t002:** *t*-test results for learning achievement, learning anxiety, critical thinking, and learning self-efficacy.

Variable	Groups	*N*	Mean	SD	*t*	*d*
Learning achievement	Experimental group	23	4.74	0.31	1.77 *	0.53
	Control group	24	4.53	0.36		
Learning anxiety	Experimental group	23	4.74	0.31	1.87 *	0.60
	Control group	24	4.53	0.36		
Critical thinking	Experimental group	23	4.74	0.31	1.57 *	0.38
	Control group	24	4.53	0.36		
Learning self-efficacy	Experimental group	23	4.74	0.31	1.80 *	0.57
	Control group	24	4.53	0.36		

* *p* < 0.05.

## Data Availability

We do not have the right to share the hospital’s data.

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
