# Peer review of "Fostering Nursing Staff Competence in Personal Protective Equipment Education during COVID-19: A Mobile-Video Online Learning Approach"

_ijerph, 2022, doi:10.3390/ijerph19159238_

Round 1
Reviewer 1 Report
Manuscript ID: IJERPH-1780144, entitled “Efficacy of Personal Protective Equipment Education in Nursing Workplace”
I wish the thank the editors for the opportunity to review “Efficacy of Personal Protective Equipment Education in Nursing Workplace”. I applaud the authors for their work, however, there are some issues needed to be clarified in the presentation in this research.
Title should be revised to match the content.
Abstract
1. Abstract should be followed the format of IJERPH, and revised to make clear. For example, “for most student,…” “this study proposes… to help nursing students…” “. It’s a little confusing that this study was conducted with nursing staffs.
2. The Conclusions of Abstract should be re-organized. Some findings should be in the Results of Abstract.
3. This study was conducted with 47 nursing staff, divided into two group. …The study involved a pre- and post-test examining…However, the results of this study did not show the pre- and post-test in each group.
Introduction
1. The introduction should be re-organized and integrated to make it clearer and more readable. For example, the authors pointed out the teaching switch to online teaching due to COVID-19. If this is an important view, please revise the title. In addition, the content of Introduction should be re-organized to follow Title.
2. The importance of this study should be emphasized in the Introduction section.
3. The last sentence of Introduction could be integrated into Aim. “Therefore, the purpose of this study…”
4. The reference format should follow the Journal.
Aim
This part could be re-organized with the last sentence of Introduction. In addition, the aim of this study should be revised to make clear to follow the Method.
Methods
1. Methods could be included “Study Design, Study Settings and Participants, Intervention, Instruments, Statistical Methods, Study Process”. Please follow the CONSORT GUIDELINES.
2. The content of Methods should be re-organized. For example, how to recruit the participants, how to allocate into experimental group and control group, how about the sample size…?
3. The intervention should be stated more clear, and should be included how to do in the experimental group and control group.
4. Instruments, How to measure the learning achievement, learning anxiety, critical thinking, and self‐efficacy as well as demographic data should be describe more details.
5. How about the study ethics? Did this study approve by IRB?
Results
1. The content of Results should be consistent with Table 1. In addition, are there any difference of demographic data between experimental group and control group? There are also some missing message in Table 1.
2. “In contrast, the scores of the experimental group were significantly higher than those of the control group for all four factors after they had received their 190 training (t = 2.64–10.46, p < 0.5). Thus the MVOLA strategy significantly reduced the participants’ learning anxiety and improved their learning achievement, critical thinking, and self‐efficacy.” Please show these results in Table.
Discussion
1. Discussion should be followed the main findings and discussed with the literatures.
Limitations could be changed to “Limitations and Recommendations”
Conclusions should be re-organized and concise.
The reference citation should be followed by this journal’s format.
Author Response
Dear Editor and Reviewers,
It is my pleasure to re-submit our revised manuscript (ijerph-1780144). According to the reviewers’ suggestion, the article title has been corrected to “Fostering nursing staff competence in personal protective equipment education during COVID-19: A mobile-video online learning approach”. In this revised version, the revised manuscript and the point-to-point responses have been resubmitted. This revised
version has been proofread by a professional native English speaker. All the changes made according to the comments are marked in red for further reference.
Thank you for your valuable comments.
Sincerely,
Authors.

Reviewer 2 Report
Personal protective equipment (PPE) training is important. The article in this field is a special viewpoint.
・In the abstract, the authors described 'how to wear PPE... and acquire related knowledge...' However, assessment questions used a learning-anxiety scale. Please correct the aim and method in the present study, as consistent.
・In the part of Methods, particpants' facilities should be cralify.
・Conclusion is too long in abstract and body text. Please describe it more simple and based on the results.
Author Response

(The authors gave the same response as above.)

Reviewer 3 Report
Comment on ijerph-1780144
1. The introduction part could be written in a concise and academic manner. First, go directly to your research theme (MOVLA or PPE, you need to make it clear). Second, give a short review of what has been done in this area. Third, point out some research gaps or weaknesses in the area (e.g., what has not been done in the area). Fourth, propose your research purpose (you have done it, but there is minor difference in your statement. On page 3, “Therefore, the purpose of the study is that ---“. “2. Aim. The purpose of the study is to ---“. Finally, briefly articulate the contribution of the study.
2. Add a part of literature review to replace “2. Aim”. In this part, you could move some review in the Introduction part and Discussion part here. More importantly, you need to provide strong theoretical argument for your major proposals. For example, why can using an MVOLA reduce the participants’ learning anxiety, and improve their learning achievement, critical thinking skills, self‐efficacy, and ability to don and doff PPE correctly?
3. In your Methods part, cite a couple of sample or reference papers used for your data analysis.
4. In Results part, please report descriptive statistics and correlations.
5. Reorganize your Discussion and Conclusion parts. (1) Major findings. (2) Theoretical implications. (3) Practical implications. (4) Limitations and direction for future research. (5) Concise conclusion. Move some discussions in Conclusion part to Theoretical implications. While drafting your theoretical implications, you may use the following style. “The study has three important theoretical implications: First, ---. Second, ---. Third, ---“.
Author Response

(The authors gave the same response as above.)

Round 2
Reviewer 1 Report
Dear authors,
Thank you very much for your efforts. I just have two suggestions. 1)Some format should be checked in the "Intervention" section. 2)The conclusions should be concise without reference.
Author Response
Dear Editor and Reviewers,
It is my pleasure to resubmit our pending minor revisions manuscript (ijerph-1780144 R2). According to the reviewers’ suggestion, the revised manuscript and the point-to-point responses have been resubmitted. A professional native English speaker has also proofread this revised version. All the changes made according to the comments are marked in red.
Thank you.
Sincerely,
Authors.

Reviewer 2 Report
Authors revised the manuscript based on the comment.
Author Response

(The authors gave the same response as above.)
